# Snow Surface Roughness across Spatio-Temporal Scales

**Steven R. Fassnacht** [1,2,3,*] **, Kazuyoshi Suzuki** [4] **, Jessica E. Sanow** [1] **, Graham A. Sexstone** [5] **, Anna K. D. Pfohl** [1] **, Molly E. Tedesche** [6,7] **, Bradley M. Simms** [1] **and Eric S. Thomas** [1]

1. ESS-Watershed Science, Colorado State University, Fort Collins, CO 80523-1476, USA; jessica.sanow@colostate.edu (J.E.S.); anna.pfohl@colostate.edu (A.K.D.P.)
2. Cooperative Institute for Research in the Atmosphere, Fort Collins, CO 80523-1375, USA
3. Natural Resources Ecology Laboratory, Fort Collins, CO 80523-1499, USA
4. Japan Agency for Marine-Earth Science and Technology (JAMSTEC), 3173-25 Showamachi, Kanazawa-Ku, Yokohama 236-0001, Kanagawa, Japan; skazu@jamstec.go.jp
5. U.S. Geological Survey, Colorado Water Science Center, Denver Federal Center, P.O. Box 25046, MS-415, Denver, CO 80225-0046, USA; sexstone@usgs.gov
6. Cold Regions Research & Engineering Laboratory, US Army Corps Engineer Research & Development Center, 72 Lyme Rd., Hanover, NH 03755, USA; metedesche@alaska.edu
7. Institute of Northern Engineering, University of Alaska Fairbanks, 1764 Tanana Loop, Fairbanks, AK 99775-5910, USA
* Correspondence: steven.fassnacht@colostate.edu; Tel.: +1-0970-491-5454

**Abstract:** The snow surface is at the interface between the atmosphere and Earth. The surface of the snowpack changes due to its interaction with precipitation, wind, humidity, short- and long-wave radiation, underlying terrain characteristics, and land cover. These connections create a dynamic snow surface that impacts the energy and mass balance of the snowpack, blowing snow potential, and other snowpack processes. Despite this, the snow surface is generally considered a constant parameter in many Earth system models. Data from the National Aeronautics and Space Administration (NASA) Cold Land Processes Experiment (CLPX) collected in 2002 and 2003 across northern Colorado were used to investigate the spatial and temporal variability of snow surface roughness. The random roughness (RR) and fractal dimension (D) metrics used in this investigation are well correlated. However, roughness is not correlated across scales, computed here from snow roughness boards at a millimeter resolution and airborne lidar at a meter resolution. Process scale differences were found based on land cover at each of the two measurement scales, as appraised through measurements in the forest and alpine.

**Keywords:** snowpack properties; random roughness; fractal dimension

## 1. Introduction

During periods of snow cover, the snowpack is the interface between the atmospheric boundary layer and the land surface [1]. The geometry of the snowpack surface controls atmosphere–snow heat transfer [2], which influences global climate [3], impacts water resource availability through surface sublimation [4], and dictates blowing snow dynamics and accumulation patterns [5]. Most models consider snow surface roughness to be a constant parameter for snow on the ground [6], yet in reality, snow surface roughness is highly variable [6–9]. It has been noted that for snow cover on the canopy, roughness length has been presented as a function of the amount of snow cover [10]. Using snow surface roughness as a constant can impact model estimates of energy exchange over a season and consequently peak SWE, compared to using dynamic snow surface roughness; as such, this paper provides process-based insight to inform physically based snow models to more accurately resolve snow water resource availability. The texture of the snowpack surface can be measured using a variety of methods [2,11–14]. Geometric snow surface data can be used to estimate roughness metrics, including aerodynamic roughness length

($z_0$) [2,6,15], random roughness (RR) [11,16], and fractal dimension (D) [11,17]. In this study, we used RR and D to assess the dynamic nature of snow surface roughness from various surface geometry measurements over a seasonal snowpack environment. We address the following research questions: (1) Is snow depth variability consistent over time? (2) Does snow surface roughness vary over space? (3) Does snow surface roughness vary over time? (4) Does snow surface roughness vary as a function of land cover? (5) Does snow surface roughness vary across spatial scales, i.e., at different resolutions? and (6) Are roughness metrics (i.e., RR and D) correlated?

## 2. Study Sites

The field data used in this study are from the National Aeronautics and Space Administration (NASA) Cold Land Processes Experiment (CLPX), which were collected in the northern Colorado Rocky Mountains (between 40 and 40.5 degrees north and 105.8 and 106.8 degrees west) during four Intensive Observation Periods (IOPs) in late February and March of 2002 (IOP1, IOP2) and 2003 (IOP3, IOP4) [18]. This includes airborne light detection and ranging (lidar), collected during the last IOP on 25 March 2003 [19]. For this paper, four 1 km² intensive study areas (ISAs) were selected (Figure 1), each comprising one hundred, 100 m × 100 m grid cell areas. The Fraser Alpine (FA) ISA was chosen because it is a mixture of forest and alpine, consisting of 31% forest, 51% alpine, and 18% at treeline or krummholz (Figure 1). The other three ISAs contain a variety of other land cover types: the Fraser Fool Creek (FF) ISA has remnants of patch cutting illustrated by a mosaic of forested and regrowth, the Rabbit Ears Spring Creek (RS) ISA is primarily covered by meadows and deciduous forest (Aspen, *Populus tremuloides*), and the Rabbit Ears Walton Creek (RW) ISA consists of primarily meadows with some clumps of Spruce–Fir forest (*Picea engelmannii* and *Abies lasiocarpa*). Several other ISAs established during the 2002–2003 CLPX campaign were not considered in our study, including the Fraser St. Louis Creek (FS) ISA, because it represents a uniform forest. Additionally, the Rabbit Ears Buffalo Pass (RB) ISA was not selected, since the roughness board images were difficult to process for most IOPs due to snowfall occurring during data collection. The three North Park ISAs were also not considered, as they have a shallow snowpack [20], and roughness board insertion was difficult at many points.

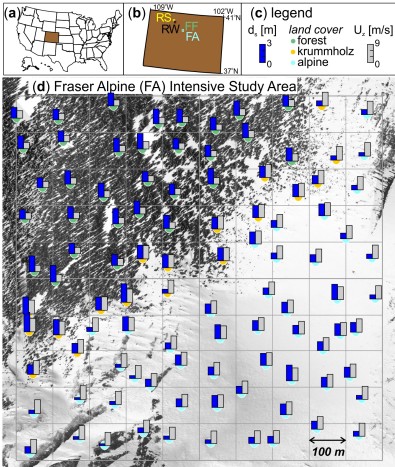

**Figure 1.** The CLPX study sites within (**a**) the United States, (**b**) the state of Colorado (FF = Fraser Fool Creek, RS = Rabbit Ears Spring Creek, RW = Rabbit Ears Walton Creek), (**c**) the legend, (**d**) highlighting the Fraser Alpine (FA) Intensive Study Area, which consists of 31% coniferous forest in the northwest, 51% alpine in the southwest through northeast, and 18% krummholz in between forest and alpine. The snow roughness board data [18,21] are for each of the locations shown, while the lidar data [19,22] are for each of the 100 m × 100 m grid cells. Wind data obtained from Sexstone et al. [23,24].

## 3. Data

Roughness estimates from 1 m long snow boards at approximately one millimeter resolution were collected by inserting a black ABS plastic board vertically into the snowpack that was subsequently photographed with a digital camera [18]. For the FA ISA, two of the March 2003 snow depths were shallower than 10 cm, and the snow surface roughness was measured at the second snow depth measurement location within the same 100 m × 100 m extent [18]. During the last sampling period, lidar was used to measure the snow surface, and those data were compared to the roughness boards. Point snow depth data were co-located with the snow roughness measurements. These were measured to the nearest 1 cm using a depth probe [18]. The snow board [21], lidar [22], and snow depth [25] data are available from the National Snow and Ice Data Center <https://nsidc.org> (last accessed 16 April 2023).

Wind speed estimates over each 100 m × 100 m area matching the CLPX extents were obtained for the FA site (data from Sexstone et al. [23,24]). Wind speed estimates were based on 1/8th-degree grid spacing North American Land Data Assimilation System (NLDAS-2) reanalysis forcing data [26] that were downscaled to a 100 m spatial resolution using MicroMet [27], a high-resolution meteorological distribution model, as described in Sexstone et al. [23]. The 100 m × 100 m wind speed estimates included adjustments to account for the presence of forest canopies, following Liston and Elder [28]. The maximum and mean daily winter wind speeds over the surface when snow covered were estimated for the period 2011–2015 [23]. The wind rose from the midpoint of each ISA illustrated that the wind mostly blew from the west [17]. For FA, land cover as forest, alpine, or krummholz/treeline was determined using visual observation of the aerial photograph (Figure 1d) and the mean January through March wind speed (Figure 2).

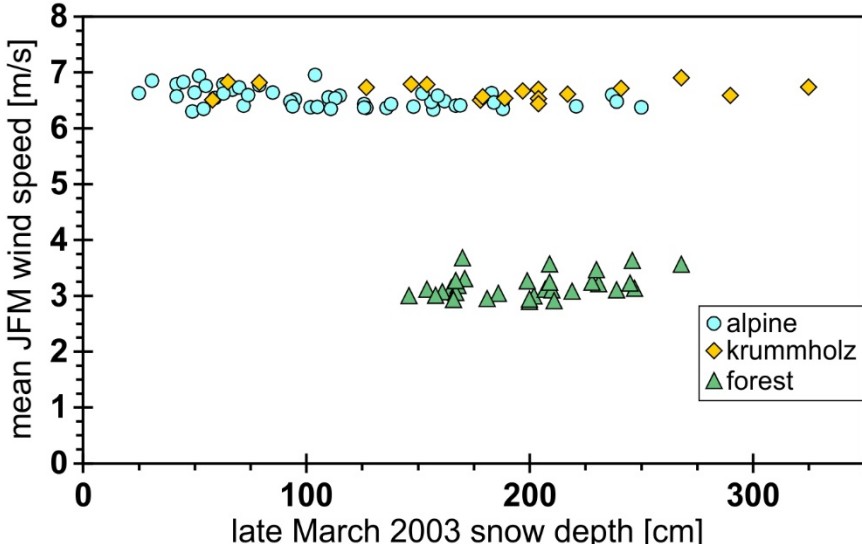

**Figure 2.** Mean January through March 100 m pixel wind speed across each grid cell of the FA ISA from Figure 1d, divided into alpine, krummholz, and forest pixels. Snow depth data are from the data of Elder et al. [18,25]. Wind data are for the years 2011–2015 from Sexstone et al. [23,24].

## 4. Methods

### 4.1. Digital Image Analysis and Lidar Processing

The image analysis technique developed by Fassnacht et al. [11] was used to convert raw snow surface roughness board images into a detrended series of X, Y coordinates. Each of the 100 photographs per ISA and IOP was cropped to fit the board extent. Any noise, such as snowfall, was manually removed by masking the specific area black. The cropped and masked images were converted into ASCII text files of digital numbers. A digital number threshold was established (Fassnacht et al. [11] used 140) to distinguish white

snow versus the black board. Finally, the snow surface was detrended using a best-fit line to remove biases, in this case from photographs taken on an angle and/or a non-smooth snow surface [11].

Two-dimensional surface data were obtained from airborne lidar [19] and are available from the National Snow and Ice Data Center [22]. Specifically, the ~1.5 m resolution lidar data were interpolated to a 2 m resolution over the study domain using ordinary kriging [13]. For each 100 m × 100 m extent, the lidar surface was detrended with a two-dimensional plane to remove elevation-based bias in the roughness computations [13].

### 4.2. Random Roughness

RR is a roughness metric that quantifies the intensity of surface roughness as one number [11,16]. It does not consider the spatial structure, i.e., the relative location of roughness elements [17]. RR is the standard deviation of elevations from the mean surface; as such, larger values represent a rougher snow surface. Each curve (board) or surface (lidar) was detrended to remove bias [11].

### 4.3. Fractal Analysis

The value of D describes the nature of the snow surface; a value of 1 is a line, a value of 2 is a plane, and a value of 3 is a surface. For the roughness boards, D is between 1 and 2 such that when D is close to 1, the surface is well organized, whereas a D close to 2 is approaching a random. For the lidar data, the value of D is 1 more, i.e., between 2 and 3. It is derived from the slope of the best-fit power function from variogram analysis [11,17], and it identifies scaling processes with similar values. For each roughness board or lidar area, the semi-variance was plotted as the variance between surface elevation points of equal distance. The lag distance is the average distance between measurements. Log-log space was used, and the power function was fit to the data starting at the shortest lag distance until the scale break (SB) or change in slope (end to a similar scaling process). The value of D was computed as 3 minus the power function exponent divided by 2 [17]. More information and examples are provided in Appendix A. A higher slope is more organized and thus has a lower fractal dimension.

### 4.4. Data Analysis

The distributions of snow depth and RR were plotted for each ISA and each IOP, as well as for each of the three land cover types (see Figure 1d) for the FA ISA. The value of D was only computed for the FA ISA since it was the only ISA with different land cover types (Figure 1d). The temporal correlation coefficient (R) was computed between each snow depth, RR, and D value over the four IOPs. To quantify snow surface roughness across scales, lidar- and board-derived metrics were compared for IOP4 at FA. Specifically, RR and D were compared both across scales at the two resolutions and versus one another (D versus RR). Since RR has units of length, it was standardized by dividing by the resolution to further quantify the correlation of the derived roughness metrics.

## 5. Results

For FA IOP4 (late March 2003), there was no correlation between snow depth and wind speed (Figure 2). The mean winter wind speeds were similar among the alpine and krummholz land cover pixels (6–7 m/s). Among the forest, they were approximately half of the other two land cover types (3–4 m/s) (Figure 2). Wind speeds varied from year to year (not shown), and the maximum wind speeds were highly correlated with the mean wind speed.

Near the Fraser ISAs (FA and FF), 2002 was a lower-than-average snow year (Figure 3). From the snow telemetry data (Figure 3) and the CLPX point depth measurements (Figure 4a), IOP1 and IOP2 had similar snow depths. Snow year 2003 was about average until a large snow event across the northern Front Range of Colorado in mid-March, as seen at the two Fraser sites (FA and FF); IOP4 occurred 5–10 days after this event (Figure 3). On average,

the snow depth was the same at the Rabbit Ears sites (RS and RW) in 2003 (Figure 4a). The point snow depth measurements were very consistent between measurement dates at the FA forest (R > 0.9) and krummholz (R > 0.92) locations (Figure 4b) but less so in the alpine (R from 0.65 to 0.81). Snow depth was least correlated at RS (R was as low as 0.32 between IOP2 and IOP3). FA alpine had the least mean amount of snow (Figure 4a).

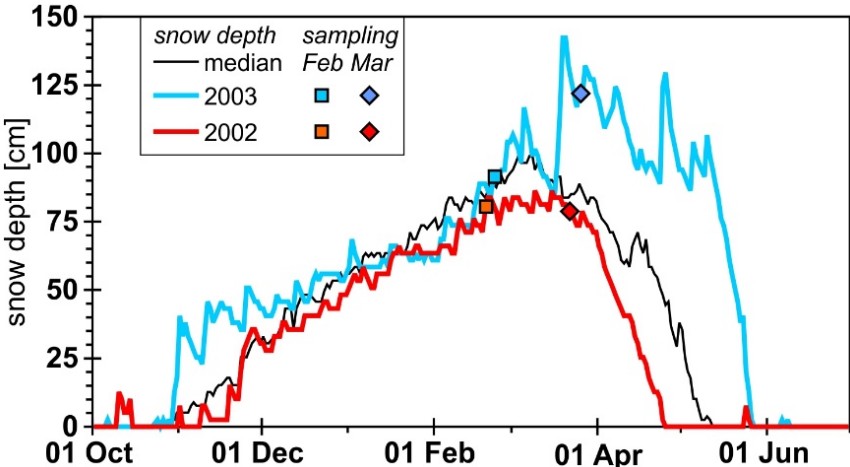

**Figure 3.** Snow depth time series from the Middle Fork Camp (station 1014) snow telemetry station (from https://www.nrcs.usda.gov/wps/portal/wcc/home/; last accessed 17 May 2023) near the Fraser sites for the 2002 and 2003 winters, illustrating the dates of CLPX data collection in late Feb and late Mar [18].

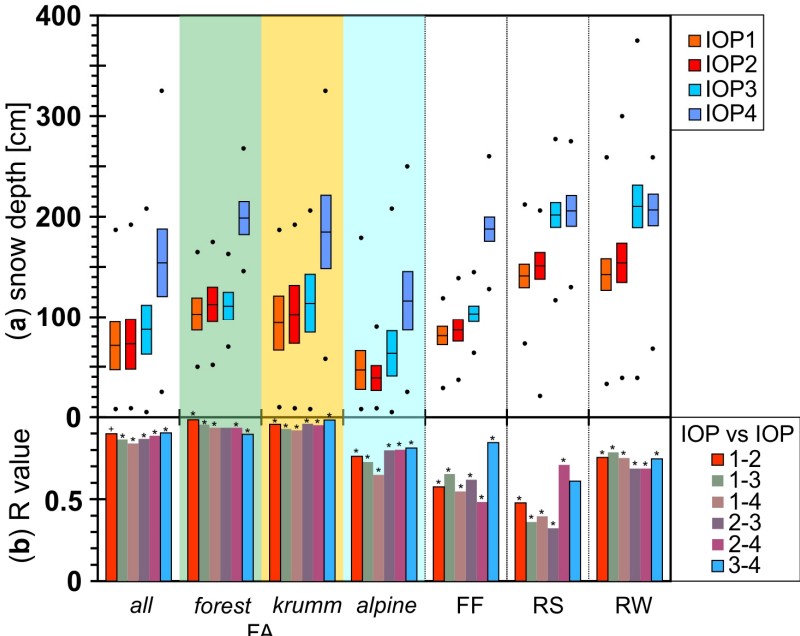

**Figure 4.** (**a**) Snow depth [25] distribution at the location where the snow board photo [21] was taken during the four IOPs (2002 and 2003 winters in late February and late March) at the four ISAs (FA = Fraser Alpine, FF = Fraser Fool Creek, RS = Rabbit Ears Spring Creek, RW = Rabbit Ears Walton Creek). The FA data were divided into forest, krummholz (krumm), and alpine. The line in each box is the mean snow depth, the box represents +/− one half standard deviation, and the dots represent the maximum and minimum. (**b**) The correlation coefficient (R) between snow depths collected at the same locations from two different time periods (i.e., a pair of IOPs). Correlations (**b**) denoted by * and + are statistically significant at the $p < 0.05$ and $p < 0.1$ levels, respectively.

When snow board data were averaged by ISA for each IOP, the mean RR values were similar (i.e., ~2.9–5.6 mm). However, variability in RR values was high at each site and observational period, often spanning ~2.5 orders of magnitude (i.e., 0.6–20 mm). At FA, the mean snow board RR was greatest during the last sampling date for the forest and alpine, and least on that date for the krummholz (Figure 5a). The RR values were poorly correlated between dates (Figure 5b), with the best correlation occurring between IOP1 and other dates in the FA forest (R of 0.32–0.38).

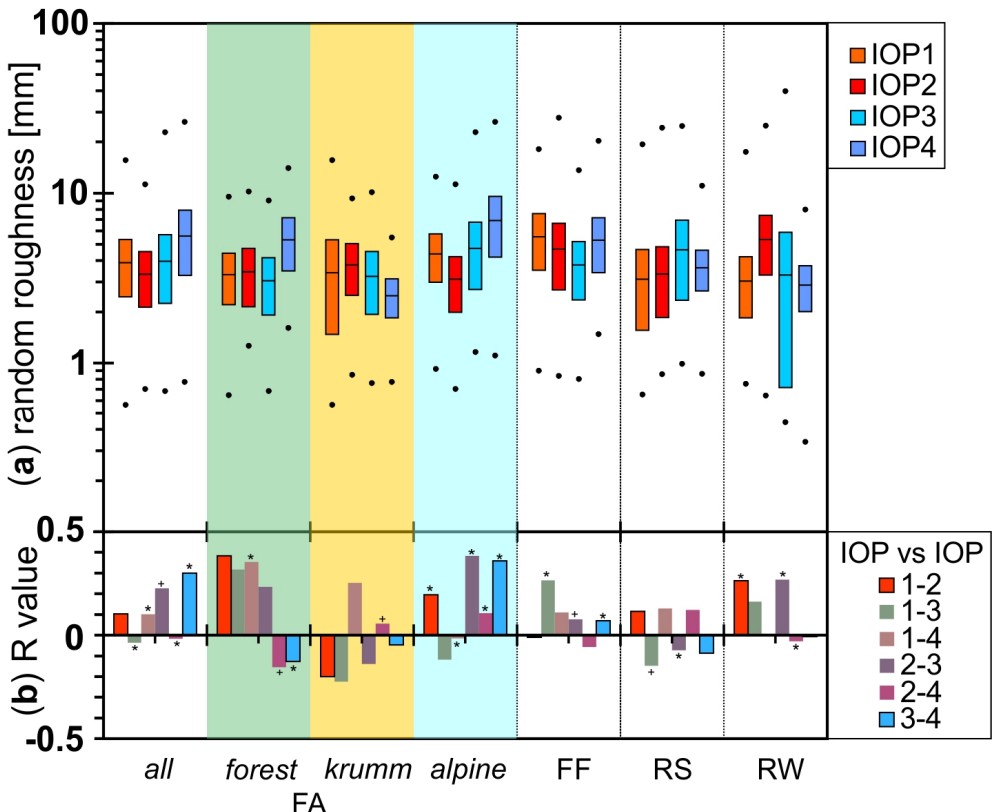

**Figure 5.** Snow board [21] (**a**) random roughness (RR) during the four IOPs (2002 and 2003 winters in late February and late March) at the four ISAs (FA = Fraser Alpine, FF = Fraser Fool Creek, RS = Rabbit Ears Spring Creek, RW = Rabbit Ears Walton Creek). The FA data were divided into forest, krummholz (krumm), and alpine. The line in each box is the mean RR, the box represents +/− one half standard deviation, and the dots represent the maximum and minimum. A logarithmic scale is used to highlight the RR differences for the small values. (**b**) The correlation coefficient (R) value between RR collected at the same locations from two different time periods (i.e., a pair of IOPs). Correlations (**b**) denoted by * and + are statistically significant at the $p < 0.05$ and $p < 0.1$ levels, respectively.

The snow board D varied from date to date, with the second IOP having the largest average D, except for in the krummholz (Figure 6a). The range of D was from 1.1 (organized) to 1.95 (almost fully random). The alpine had the lowest D values, with the range of D values being similar between forested and krummholz. Across FA and among the different land covers, the mean D was lowest on the last sampling date. It was poorly correlated between dates (Figure 6b), with little correlation in the forest (R~0.2) and essentially no correlation in the krummholz and alpine.

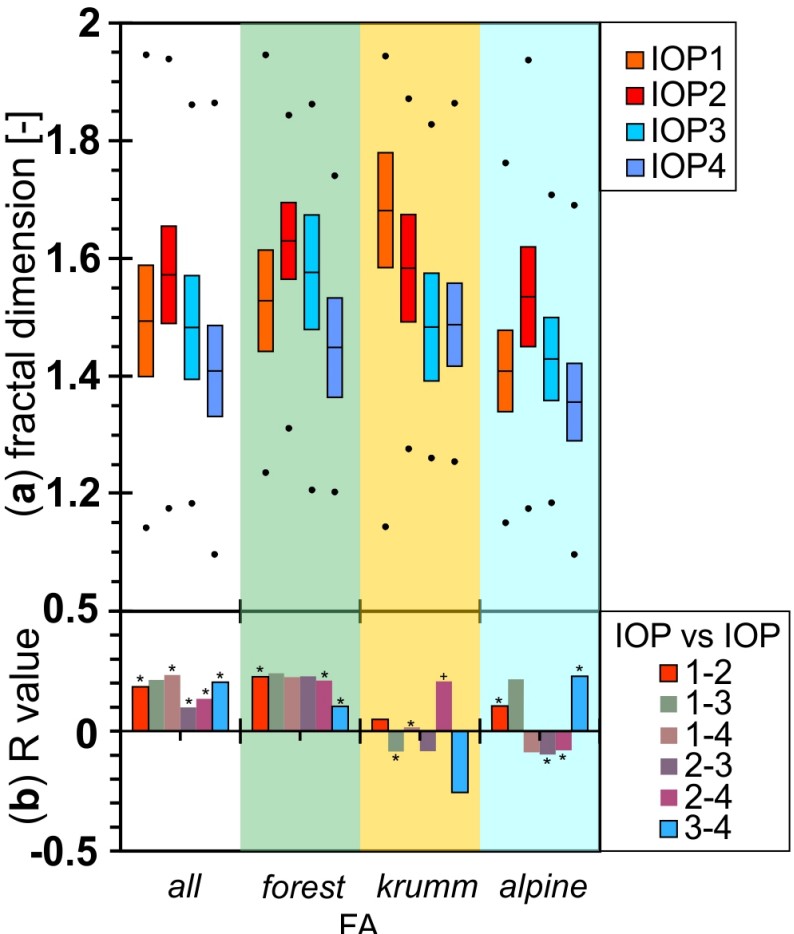

**Figure 6.** Snow board [21] (**a**) fractal dimension (D) during the four IOPs (2002 and 2003 winters in late February and late March) at the Fraser Alpine (FA) ISA, which was divided into forest, krummholz (krumm), and alpine. The line in each box is the mean D, the box represents +/− one half standard deviation, and the dots represent the maximum and minimum. (**b**) The correlation coefficient (R) D estimated at the same locations from two different time periods (i.e., a pair of IOPs). Correlations (**b**) denoted by * and + are statistically significant at the $p < 0.05$ and $p < 0.1$ levels, respectively.

There was no correlation among RR (Figure 7a) or D (Figure 7c) across scales, i.e., from the snow roughness board to the lidar. For the lidar and the boards, RR varied by up to a factor of 20 (Figure 7b). The range of D for lidar (range of ~0.3, values from 2.1 to 2.4) was approximately half that of the boards (range of ~0.6, values from 1.1 to 1.7) (Figure 7c,d).

The correlation between D and RR was consistent across the four IOPs, especially when comparing the four different land cover types (Figure 8a–d). For the alpine, the D and RR values were smaller than the forested and krummholz land cover types. Smaller D and RR values were most apparent in the lidar datasets, which also show a weaker correlation between D and RR values (i.e., $R^2$ of 0.22 versus 0.58, respectively) for the entire domain during IOP4 (Figure 8e).

By reducing the lidar D by a factor of 1 ($D_{adj}$), (i.e., changing the range from 2–3 to 1–2, meaning plane-surface to line-plane) and then standardizing RR by pixel size ($RR_{std}$), we see more distinctive correlations (Figure 9), as compared to the raw values (Figure 8). The $D_{adj}$ and $RR_{std}$ values have higher variance in regression residuals for board-derived versus lidar-derived values (Figure 9), and the $R^2$ values are also higher for forest and alpine (Figure 9b,d) for the boards versus the lidar. The opposite is seen for the krummholz (Figure 9c), due to fewer points (18) and two extreme values for the boards (D from 1.3 to 1.9 for a similar RR value). The alpine has the lowest lidar RR values and the highest board RR values (Figure 9d).

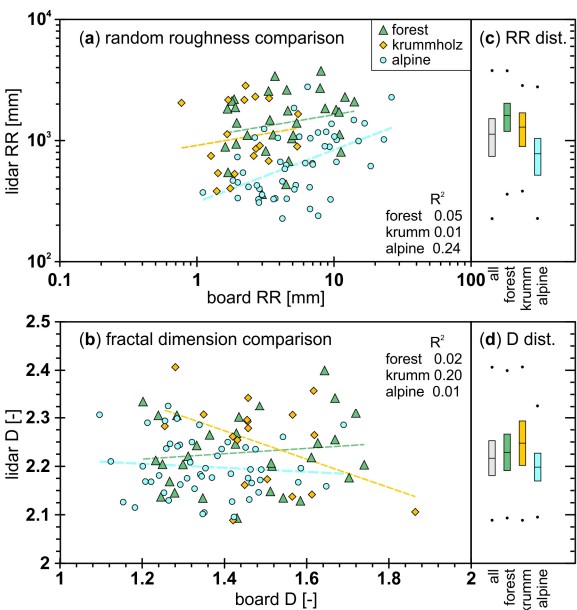

**Figure 7.** Lidar [22] pixel versus snow roughness board [21] comparison of (**a**) random roughness (RR) and (**b**) fractal dimension (D) for IOP4 (late March 2003). The best-fit function and the coefficient of determination ($R^2$) for the three land cover groups is shown, and all correlations are statistically significant at the $p < 0.05$ level. The distribution of lidar pixels is shown in (**c**) RR and (**d**) D (see Figure 6 for the snow roughness boards).

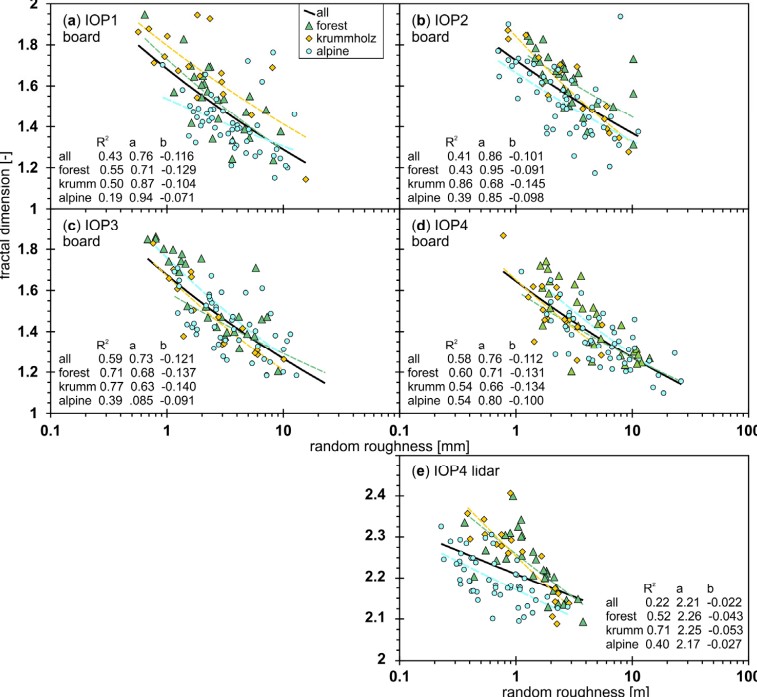

**Figure 8.** Snow roughness board [21] fractal dimension (D) versus random roughness (RR) for (**a**) IOP1, (**b**) IOP2, (**c**) IOP3, and (**d**) IOP4 at the Fraser Alpine ISA divided by forest, krummholz, and alpine land cover. (**e**) Lidar-based [22] IOP4 D versus RR (with different scales). The coefficient of determination ($R^2$), coefficient (a), and exponent (b) for the power function fit to all data, and the three land cover groups are listed. The line shown is the best-fit power function for all the data. All functions were statistically significant at the $p < 0.05$ level.

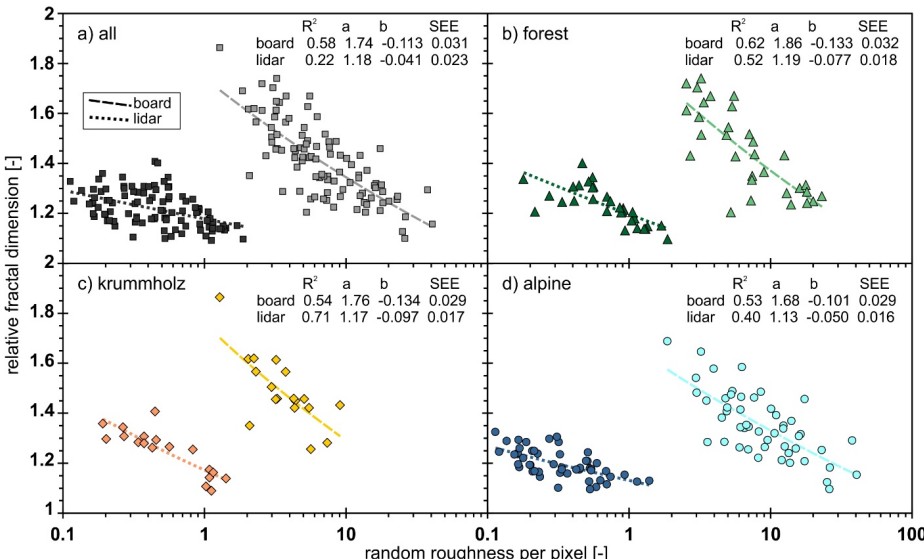

**Figure 9.** Correlation between relative fractal dimension (scaled to between 1 and 2) and random roughness standardized by resolution for the lidar [22] and board [21] estimates of IOP4 (late March 2003) across (**a**) the entire domain, (**b**) forest, (**c**) krummholz, and (**d**) alpine, including the statistics for the best-fit power function: the coefficient of determination, coefficient (a), exponent (b), and standard error (SE). All functions were statistically significant at the $p < 0.01$ level.

## 6. Discussion

Snow depth distribution patterns have been shown to be consistent inter-annually [29–31] and seasonally [32]. Therefore, snow depth is also assumed to be spatially consistent intra-annually [33]. From our analysis, the intra-annual consistency is slightly greater (R increase of 0.05) than the inter-annual consistency (Figure 4b), with snow depth being more consistent in 2003 (IOP3 and 4) than in 2002 (IOP1 and 2). This is likely due to the substantial March snowfall that was observed at FA and FF (Figure 3). The snow depths used herein are from manual depth probe measurements [18], which have inherent biases due to over-probing [34] and location uncertainty [34–36]. More efficient data collection methods could enable a better assessment of intra- and inter-annual variability, such as automated depth probes [34], various forms of lidar (e.g., Deems et al. [37]), and photogrammetry techniques (e.g., Nolan et al. [38]). These advanced data collection methods could be especially impactful when implemented more frequently over the same period or winter season (e.g., Pflug and Lundquist [39]).

Since snowpack surfaces are continuously changing and evolving, snow surface roughness varies spatially (Figures 5a and 6a) [7,8] and temporally (Figures 5b and 6b) [6,8], as it is driven by interactions with meteorological forces. The observed spatio-temporal variability from the boards is partially due to the fine resolution (<1 mm) and small sampling extent (1 m); this scale can identify very local features that are linked to snow depth characteristics [8]. Temporal variability is correlated with the timing of the sampling relative to fresh snowfall events (Figure 3) [40,41]. Thus, it is relevant to track the snow surface roughness evolution with respect to what is occurring to the snowpack (accumulation, compaction, ablation, etc.). At the lidar scale (2 m resolution at the 100 m extent in this study), terrain characteristics dictate spatial variability [13]. Temporal variations at coarser lidar resolutions were not evaluated in our study; however, such investigations have been conducted by others, such as with the Airborne Snow Observatory dataset for the Tuolumne Basin in California, USA, as illustrated in Pflug and Lundquist [39].

In our study, the snow surface roughness in the krummholz was similar to the forest, with RR being greater (Figure 5a) and D being lesser (Figure 6a) than in the alpine. In the forest, snow interception dominates the snow surface characteristics [10] and dictates the correlation length (8–15 m) [17,42], whereas in the alpine, blowing snow dominates and

has a much longer correlation length for measurements (40+ meters) [17,42]. Although the wind characteristics in the krummholz were the same as in the alpine (Figure 2), the process behavior between forested, including krummholz, versus non-forested (alpine) partially dictated snow surface roughness characteristics. Specifically, in the forest or krummholz, canopy interception is an important process, whereas in the alpine, shrub–snow interaction is relevant [20,43], and blowing snow dominates [23].

The differences between snow surface roughness in the forest, alpine, and krummholz did vary somewhat over time (Figure 8). Due to the increased wind in the krummholz, there can be a large amount of blowing snow, causing large snow depth variability (Figure 4a) [44]. For the krummholz, there is less snow accumulation below trees during snowfall if winds are low, but subsequent blowing snow fills in much of this snow under individual trees [45]. Snow surface roughness varies as a function of land cover characteristics (Figures 5, 6, and 8). This has the potential to impact how moisture and wind regimes drive sublimation [46].

At the lidar resolution, these differences, i.e., spread between krummholz and forest versus alpine (Figure 8e), were more distinct than for the boards. Also, the "slope", as represented by the b values (i.e., the exponent of the best-fit power function), is very similar between forest and krummholz (Figure 9). We see a gap between measurement scales over the three orders of magnitude of resolution (Figure 9). For example, the snow boards' spatial extent is too small to capture snow drifts, whereas the airborne lidar data are too coarse to capture the details of the spatially dynamic nature of snow drifts. At finer resolution, the snow surface characteristics are strongly related to snow depth, whereas at coarser resolution, the snow surface is more related to the ground surface characteristics [8]. Overall, the correlation between D and RR is lowest for the alpine, as represented by the $R^2$ values (Figures 8 and 9). We do see correlations between the two scales, with more of these correlations occurring in forest and krummholz land types than in the alpine (Figure 9), impacted by wind dominance (alpine) compared to canopy dominance (forest) or both (krummholz). This yields differences in the scales [17,42].

To better assess how snow surface roughness changes across scales, we could measure the snow surface (spatially and temporally) at the centimeter scale over an extent of 10s of meters [8]. This would be a finer resolution than suggested by Andreas [12], who used manual 0.5 m sampling over 128 m transects via spectral analysis. Although terrestrial lidar is often employed at coarser resolutions [37,47], especially over longer distances [48], it can also be used over smaller spatial domains and at finer resolutions. New technology, such as drones or uncrewed aerial vehicles (UAVs), can be used with photogrammetry or structure from motion [49], with newer, smaller lidar units becoming practical for drones [50]. Finally, new lightweight hand-held lidar tools are starting to be useful for small domains [51].

## 7. Implications

There are differences between snow surface roughness expressed as RR versus D (Figure 8), but there is more correlation between these metrics for the boards (Figure 8a–d) than lidar (Figure 8e). Numerous surface roughness metrics exist [14], but relatively few are used for snow due to the complexity of snow surface characteristics [11]. Values of $z_0$ can be computed using surface geometry [2,6,15] by identifying individual surface elements and computing the ratio of the cross-section area perpendicular to the wind to the horizontal area of the elements [13]. This geometric assessment may be performed through photogrammetry or structure from motion [52]. However, surface geometry is not the only factor influencing $z_0$ [12], as wind dynamics are relevant, as they, in turn, shape the snow surface [5,44]. Overall, the variability in snow surface roughness has implications for the energy balance of the snowpack, especially the sensible and latent heat fluxes [53], plus the mass balance due to sublimation [4,54], blowing snow [5,23], and other components [55], such as net short- and long-wave radiation.

## 8. Conclusions

From the paper's objectives, we conclude as follows:

(1) The variability in snow depth is most temporally consistent in the forest (R approaching 1 over time), slightly less in the alpine (R is approximately 0.75), and least consistent in open terrain with mixed forests (R is approximately 0.6). Snow depth is more consistent intra-annually than inter-annually.

(2) Snow surface roughness, as defined by the random roughness, varies by up to 1.5 orders of magnitude over space. Mean random roughness values vary by a factor of 2 or 3 across the various study domains. This was observed for the boards and the lidar-derived snow surfaces. The fractal dimension value varies from 1.1 to 1.95 for the boards and by less than half for the lidar-derived snow surfaces (1.1 to 2.4).

(3) Snow surface roughness from the boards is not temporally consistent; the maximum R-value for random roughness is 0.35, with most intra- and inter-annual comparisons being less than 0.2. Temporal consistency is less for the fractal dimension, with R being less than 0.2. Lidar data were only available for one time period, and thus the temporal variability was not assessed.

(4) Snow surface roughness is correlated with land cover characteristics. Alpine has a larger random roughness and is more organized (lower fractal dimension) than forest. The values for krummholz are between alpine and forest.

(5) The two snow surface roughness metrics are not correlated across spatial scales, i.e., from the boards at millimeter resolution to the lidar data at meter resolution.

(6) The roughness metrics (i.e., RR and D) are well correlated, especially when separated by land cover. The correlation is more obvious when the dimension is removed from the roughness metrics.

**Author Contributions:** Conceptualization, S.R.F., K.S., B.M.S. and E.S.T.; methodology, S.R.F., K.S. and B.M.S.; formal analysis, S.R.F., B.M.S. and G.A.S.; investigation, S.R.F., K.S. and J.E.S.; data curation, G.A.S.; writing—original draft preparation, S.R.F., K.S. and J.E.S.; writing—review and editing, S.R.F., K.S., J.E.S., G.A.S., A.K.D.P. and M.E.T.; visualization, S.R.F. and K.S.; supervision, S.R.F.; project administration, S.R.F.; funding acquisition, S.R.F. and J.E.S. All authors have read and agreed to the published version of the manuscript.

**Funding:** This paper was written while S.R.F. was on a fellowship from the Japanese Society for the Promotion of Science <https://www.jsps.go.jp/> (last accessed 2 March 2023), hosted by the Japan Agency for Marine-Earth Science and Technology <https://www.jamstec.go.jp/> (last accessed 2 March 2023). Funding for K.S. was provided by the Japan Society for the Promotion of Science (JSPS) KAKENHI grants (grant numbers 19H05668, 21H04934, and 22H03758) and the Arctic Challenge for Sustainability II (ArCS II) (Program grant number JPMXD1420318865). This research was indirectly funded by the U.S. Geological Survey National Institutes for Water Resources (U.S. Department of the Interior), grant number 2019COSANOW, project "The Dynamic Nature of Snow Surface Roughness", through the Colorado Water Center.

**Data Availability Statement:** The snow board [21], snow depth [25], and lidar [22] data are openly available at the National Snow and Ice Data Center <https://nsidc.org/> (last accessed 2 March 2023). The wind data are from Sexstone et al. [23,24] and are openly available in a U.S.G.S. data release <https://doi.org/10.5066/F75M64QQ> (last accessed 14 March 2023).

**Acknowledgments:** The Japan Agency for Marine-Earth Science and Technology provided administrative and related support during the culmination of this project. B.M.S. and E.S.T. worked on this project as part of the Vertically Integrated Projects Program at Colorado State University <https://vip.colostate.edu/> (last accessed 2 March 2023). Any use of trade, firm, or product names is for descriptive purposes only and does not imply endorsement by the U.S. Government. We thank David Ray of the U.S. Geological Survey and David Barnard of the Agricultural Research Service for reviewing a draft of this paper.

**Conflicts of Interest:** The authors declare no conflict of interest.

## Abbreviations

| | |
|---|---|
| CLPX | Cold Land Processes Experiment |
| D | fractal dimension |
| FA | Fraser Alpine ISA |
| FF | Fool Creek ISA |
| IOP | Intensive Observation Period (IOP1 = February 2002, IOP2 = March 2002, IOP3 = February 2003, IOP4 = March 2003) |
| ISA | Intensive Study Area |
| NLDAS | North American Land Data Assimilation System |
| R | correlation coefficient (−1 to +1) |
| $R^2$ | coefficient of determination (0 to 1) |
| RR | random roughness |
| RS | Rabbit Ears Spring Creek ISA |
| RW | Rabbit Ears Walton Creek ISA |

## Appendix A. Fractal Analysis

For one-dimensional (1-D) data, i.e., the roughness boards, D has a value between 1 and 2 and is computed as follows:

$$D = 2 - b/2 \tag{A1}$$

where b is the slope of the variogram in log-log space or the exponent of the best-fit power function that has the form:

$$\gamma = ax^b \tag{A2}$$

where $\gamma$ is the semi-variance, x is the lag distance from the variogram, and a and b are the best-fit coefficients, analogous to the y-intercept (a) and slope (b). For two-dimensional data, i.e., the lidar-derived snow surface, D is between 2 and 3. For a 1-D surface, a D of 1 is a line, and a D of 2 is a surface, with natural surfaces typically having a value of D greater than 1 but less than 2 (Figure A1). The examples in Figure A1 show a very organized surface (Figure A1a with D~1.05), a partially organized surface with some randomness (Figure A1b with D~1.5), and an almost completely random surface (Figure A1c with D~1.95).

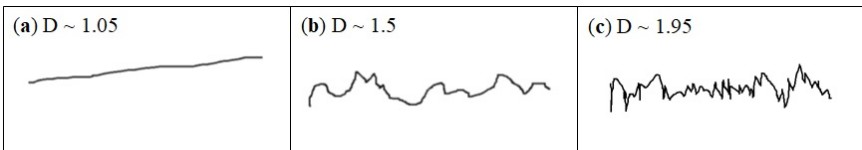

**Figure A1.** Three 1-D surface examples where (**a**) the surface is well organized, with D slightly greater than 1, (**b**) the surface is partially organized, with D approximately 1.5, and (**c**) the surface is almost completely random, with D approaching 2.

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
