# Peer review of "Snow Surface Roughness across Spatio-Temporal Scales"

_water, doi:10.3390/w15122196_

Round 1

Reviewer 1 Report

I have found the study interesting and worthy of discussion. Please consider my minor comments bellow

1- Provide coordinates for the study area

2- Too much abbreviation is used. Once reading, I had to go back and check for the abbreviation. So, I recommend reducing the number of abbrivations

3- I think methodology especially for Fractal analysis should be clarified. For example, you can provide sample Sample figures to clarify the D measure

4- I wish you have used other types of evaluation criteria such as "mutual information" or factor analysis since complex procedures most probably follow nonlinear relationships which could not be detected by "correlation coefficient"

5- I suggest providing an Appendix regarding the fractal dimension  

The quality of the text is satisfactory. But it would be better to use fewer abbreviations

Author Response

1- Provide coordinates for the study area

  • Coordinates were added to the study site map of Colorado, and are now listed in the text.

2- Too much abbreviation is used. Once reading, I had to go back and check for the abbreviation. So, I recommend reducing the number of abbreviations

  • We have a list of abbreviations at the beginning of the paper. Several abbreviations were removed including DN = digital numbers (1 occurrence), NSIDC = National Snow and Ice Data Center (1 occurrence), and SSR = snow surface roughness (17 occurrences). Eight of the remaining 13 abbreviations are from the NASA CLPX project, while four others are statistics. The last one NLDAS is used by to describe a specific, well-known dataset.

3- I think methodology especially for Fractal analysis should be clarified. For example, you can provide sample figures to clarify the D measure

  • We believe that the fractal dimension should be understood and citations are provided to explain the concept. However, we have now included information about the fractal analysis in Appendix A.

4- I wish you have used other types of evaluation criteria such as "mutual information" or factor analysis since complex procedures most probably follow nonlinear relationships which could not be detected by "correlation coefficient"

  • There are other statistics that can be used in hydrology, such as Nash-Sutcliffe Efficiency or Kling-Gupta Efficiency. The coefficient of determination used in Figures 4, 5 and 6 are appropriate since these are linear correlations and here we want to show positive versus negative correlations The correlation coefficient used in Figures 7, 8 and 9 were in log-log or log- linear space, so are appropriate. They are meant to show the strength of the correlation. Thank you for the suggestions – we will consider them in the future.

5- I suggest providing an Appendix regarding the fractal dimension  

  • We have now included information about the fractal analysis in Appendix A.

Reviewer 2 Report

I have read the manuscript with interest and it resulted me very clear in the objectives, methodology well presented, very good quality in the presented figures and, as overall, the results are easy to follow and are well supported by the analyses. Thus, I recommend the publication of the paperfor water with almost no suggested changes.

My only recommendation is about the summary section that I would reformulate, as I think that it should go enterely to the discussion section, while this section should present in the most simple language the answer to the main scientific questions stated in the introduction in a slightly more expanded way (and providing some numbers) than in the abstract

[1) Is snow depth variability consistent over time? 2) Does SSR vary over space? 3) Does SSR vary over time? 4) Does SSR vary as a function of land cover? 5) Does SSR vary across spatial scales, i.e., at different resolutions? and 6) Are roughness metrics (i.e., RR, D) correlated?]

Congratulations to the authors.

Author Response

My only recommendation is about the summary section that I would reformulate, as I think that it should go entirely to the discussion section, while this section should present in the most simple language the answer to the main scientific questions stated in the introduction in a slightly more expanded way (and providing some numbers) than in the abstract

[1) Is snow depth variability consistent over time? 2) Does SSR vary over space? 3) Does SSR vary over time? 4) Does SSR vary as a function of land cover? 5) Does SSR vary across spatial scales, i.e., at different resolutions? and 6) Are roughness metrics (i.e., RR, D) correlated?]

  • The existing section has been retitled “Implications,” and a new “Summary” section has been added. This new section speaks directly to the six objectives of the paper.

Reviewer 3 Report

The study provides valuable insights into the variability of snow surface roughness (SSR) and its implications for the energy and mass balance of the snowpack. The study effectively highlights the dynamic nature of the snow surface, which is often overlooked in current earth system models. The correlation analysis between random roughness (RR) and fractal dimension (D) metrics contributes to understanding SSR assessment.  Overall, this research presents essential findings that contribute to the field of snowpack characterization and its impact on various processes.

Author Response

Thank you for your comments.